# Development of a Conceptual Scheme for Controlling Tool Wear During Cutting, Based on the Interaction of Virtual Models of a Digital Twin and a Vibration Monitoring System

**DOI:** 10.3390/s24227403

**Published:** 2024-11-20

**Authors:** Lapshin Viktor, Turkin Ilya, Gvindzhiliya Valeriya, Dudinov Ilya, Gamaleev Denis

**Affiliations:** Department of Automation of Production Processes, Don State Technical University, Gagarina Str. 1, Rostov-on-Don 344090, Russia; tur805@mail.ru (T.I.); vvgvindjiliya@donstu.ru (G.V.); ilya.sandman@yandex.ru (D.I.); gamaleev.00@mail.ru (G.D.)

**Keywords:** virtual models, digital twins, neural networks, cutting dynamics, tool wear

## Abstract

This article discusses the issue of the joint use of neural network algorithms for data processing and deterministic mathematical models. The use of a new approach is proposed, to determine the discrepancy between data from a vibration monitoring system of the cutting process and the calculated data obtained by modeling mathematical models of the digital twin system of the cutting process. This approach is justified by the fact that some coordinates for the state of the cutting process cannot be measured, and the vibration signals measured by the vibration monitoring system (the vibration acceleration of the tip of the cutting tool) are subject to external disturbing influences. Both the experimental method and the Matlab 2022b simulation method were used as research methods. The experimental research method is based on the widespread use of modern analog vibration transducers, the signals from which undergo the process of digitalization and further processing in order to identify arrays of additional information required for virtual digital twin models. The results obtained allow us to formulate a new conceptual approach to the construction of systems for determining the degree of cutting tool wear, based on the joint use of computational virtual models of the digital twin system and data obtained from the vibration monitoring system of the cutting process.

## 1. Introduction

Improving the quality of metal part processing on metal cutting machines can be achieved by increasing the accuracy of positioning, as well as by reducing the vibrations of both the cutting tool and the workpiece itself, fixed in the machine spindle [1]. The quality of the surface of metal parts processed on a metal cutting machine is determined by a combination of a number of characteristics that affect the subsequent performance of this part. All these characteristics are related to the surface layer of the part, which is characterized by macro and micro deviations from a given geometric shape.

The most important factor determining the quality of the surface obtained during cutting is the degree of cutting tool wear. It is widely known that when the tool reaches critical wear, the vibration activity of the cutting tool increases sharply and, as a result, the quality of the surface obtained during cutting decreases. To avoid situations involving cutting metals with worn tools, it is necessary to accurately predict the development of tool wear when performing operations on a metal-cutting machine. Modern vibration monitoring systems allow, based on digital measuring systems, the prediction of the quality of surfaces obtained during cutting [2,3]. However, forecasting the development of wear on the cutting tool requires the development of complex mathematical models of the evolutionary dynamics of processes occurring during cutting [4]. The complexity of such models and their requirements for parametric identification are a problem, the solution to which would significantly increase the capabilities of modern metalworking systems.

One of the ways to solve this problem is to use a new digital paradigm in quality management and control systems, which has been called a digital twin [5,6,7]. In particular, the approach based on the use of intelligent models describing the complex dynamics of technological processes occurring during metal cutting is the most promising in this new field of scientific knowledge [8,9]. For example, in the works performed by teams under the leadership of Yu. AlTintasa, who is a world-renowned specialist in the field of digital counterparts for metalworking control systems, has proposed using digital counterparts to form new CNC programs that will allow parts to be processed without pre-settings and experiments [10]. That is, the issue of choosing technological processing modes, both in the process of solving current problems, and during the restructuring of the control system on a metal-cutting machine (the property of flexibility) can be solved using virtual models of a digital twin.

In the modern view, the technology of constructing digital twins, in terms of the synthesis of virtual models, is based on two paradigms; the first is based on the use of deterministic mathematical models [11], the second is based on the widespread introduction of neural networks [8,9]. The advantage of deterministic virtual models is their complexity and structural relationship with the metal cutting process itself. The main disadvantage of deterministic virtual models is the very complex nature of the cutting process, which requires the construction of very complex and cumbersome mathematical models describing the evolutionary dynamics of cutting processes [11]. Based on this analysis, one can see the formation of a dualistic contradiction in the process of forming virtual models of digital counterparts of cutting processes. The problem of dividing models into deterministic mathematical and intelligent (neural networks) models is the significant limitation of the use of digital twin technology to solve specific problems of metalworking.

One of the most important problems, the solution to which could significantly increase the efficiency of modern metalworking systems, is the problem of predicting the residual durability of cutting tools. The reason why existing methods and methods are limited in their accuracy of predicting the residual durability of a cutting tool is the complex and multifactorial nature of tool wear [12]. In general, both the cutting dynamics and the evolutionary changes in these dynamics associated with the increase in the degree of the cutting tool wear is a multifactorial, complex process, the exact description of which is almost impossible. In other words, here we are confronted with a certain “thing in itself” according to the sense used by Kant [13].

One of the most important directions in the development of digital twin technology is that of diagnosing various malfunctions; for example, in [14], the issue of generating labeled training datasets for various bearing malfunctions that would complement the limited measured data are considered. Here, the authors propose a new approach using a digital twin to solve the problem of limited measured data in the diagnosis of bearing failures. The results of the experiments conducted by the authors show an increase in the accuracy of fault diagnosis [14]. The same direction, but in a slightly different view, is presented in study [15]. Here, the authors point out the limitations of traditional fault diagnosis methods based on experimental data.

One of the most important directions in the development of digital twin technology is the direction of diagnosing various malfunctions; for example, in [14], the issues of generating labeled training datasets for various bearing malfunctions that would complement the limited measured data are considered. Here, the authors propose a new approach using a digital twin to solve the problem of limited measured data in the diagnosis of bearing failures. The results of the experiments conducted by the authors show an increase in the accuracy of fault diagnosis [14]. The same direction, but in a slightly different view, is revealed in article [15]. Here, the authors point out the limitations of traditional fault diagnosis methods based on experimental data. They note that in some critical industrial scenarios, such a dataset is not always available. It is digital twin technology, which creates a virtual representation of a physical object by reflecting its operating conditions, which makes it possible to diagnose malfunctions of technical systems or technological processes when there is insufficient data on malfunctions. The authors propose a fault diagnosis system based on digital twins using labeled simulated data and unmarked measured data [15]. The construction of a digital twin system that integrates sensor data from faulty bearings into the subspaces of virtual models in real time is presented in [16]. The authors refine the parameters of virtual models by comparing the results of digital modeling in the time domain with measured and captured signals [16].

An interesting direction in the development of digital twin technology may be found in the of synthesis of neural networks that diagnose possible failures based on the results of comparing model data and diagnostic system data [17]. The dual data transmission architecture proposed here by the authors provides more opportunities for the practical application of intelligent fault diagnosis with small sample sizes [17].

Based on the analysis, it can be seen that digital twin technology has become widespread in the diagnosis of malfunctions, including bearing malfunctions. Therefore, the obvious development of digital twin technologies is its application in diagnosing the wear of cutting tools in metalworking control systems.

Summarizing all the mentioned above problems, we can point out that the direction of digital twin development, using both of the above-mentioned approaches to the synthesis of virtual models, will be popularly in demand in the near future. What is meant here is the formation of a digital twin system structure that could use the strengths of both of these approaches. Such a structure should contain two levels; the first level associated with the use of deterministic models, and the second level having the ability to operate with data obtained by the digital twin system directly from the vibration monitoring system of the cutting process. In other words, the limitations of deterministic mathematical models of understanding the “things in themselves” of the cutting process [13] can be overcome by developing a conceptual scheme for a digital twin system divided into the following levels of decision-making: operational, a data-oriented level of wear control, and strategic, designed for planning works on a metal-cutting machine.

Based on this, we formulate the purpose of the study as the increase in the accuracy of predicting residual durability of a cutting tool, with the formation of a conceptual scheme for a control system, with control of wear using indirect informative signs obtained by the vibration monitoring system of the cutting process.

## 2. Synthesis of a Deterministic Mathematical Model of the Cutting Process, Using the Example of Metal Turning

Before constructing a mathematical model of the machining process, let us consider a structural diagram that reveals the main axes of deformation of the cutting tool and the workpiece, as well as the decomposition of reaction forces to formative movements along these axes. When forming this system of equations modeling the forces acting on the tool, it is necessary to take into account their properties, which are known from processing technology: the forces acting on the end surface of the tool depend on the area of the cut layer.

In the variant shown in Figure 1, the *x*, *y*, *z* coordinates denote the deformation movements of the tip of the cutting tool; the force *F*, which prevents the shaping movements of the tool, is decomposed along the axes of deformation into the following components: *Ff*—component in the feed direction, *Fp*—component in the radial direction, and *Fc*—tangential component (cutting force); the value ω—indicates the angular the rotation speed of the workpiece; and $V_{f},V_{c}$—the speeds set by the CNC program, feed rate and cutting speed, respectively.

We further note that the formation of tool wear area along the back face significantly affects the overall force response from the cutting process to the shaping movements of the tool. Based on these considerations, the overall force response of the machining process to the shaping movements of the tool can be represented as:(1)$$\left\{\begin{array}{c} \begin{array}{c} F_{f}=\chi_{1}F+F_{h}^{\left(x\right)}\\ F_{p}=\chi_{2}F+F_{h}^{\left(y\right)} \end{array}\\ F_{c}=\chi_{3}F+F_{h}^{\left(z\right)} \end{array}\right.,$$
where $\chi_{1},\chi_{2},\chi_{3}$ are the coefficients of decomposition of the cutting force *F* on the axis of the tool deformation; these coefficients depend on the angles indicated in Figure 1 in the tool plan (φ, φ1, α), and it is here that the geometry of the cutting plate is taken into account [4]. *F_h_*.—the composing force preventing the penetration of the cutting edge of the tool into the workpiece. The Fc component is of the greatest importance, as it most accurately reflects the cutting force itself and determines the oscillatory activity of the tool in the direction of the z-axis. The force itself, which prevents the shaping movements of the tool based on a hypothesis of its proportionality to the area of the cutting layer [10,11], has the following form:(2)$$F=pt_{p}S,$$
where *p*—the constant that determines the value for the specific chip pressure per millimeter of the area of the layer cut during cutting, for which the geometry of the cutting plate also plays a role, *t_p_*—depth of the cut layer (mm), and *S*—feed per revolution (mm).

It should be noted here that the actual cutting depth and the actual feed per revolution will depend on the vibration values of the tool tip; in the case of cutting depth, it will be determined as follows:(3)$$t_{p}=t_{p0}-y,$$
where y—the amount of deformation of the tool tip in the radial direction, and $t_{p0}$—the evaluation of the cutting depth set by the CNC program.

As for the amount of feed per revolution, it will be determined as:(4)$$S=V_{f}-x\left(t\right)+x(t-T),$$

We can note that it is a well-known fact of cutting processes that when the tool moves towards contact with the back face (main or auxiliary), a velocity-dependent increase in forces is observed $F_{h}=\{{F_{h}}^{\left(x\right)},{F_{h}}^{\left(y\right)},{F_{h}}^{\left(z\right)}{\}}^{T}$; this should, further, be taken into account when synthesizing an additional model of contact interaction on the back faces of the tool.

Taking into account the designations adopted earlier, we describe the power of irreversible transformations as follows:(5)$$N=\sqrt{(F_{f}{)}^{2}+(F_{p}{)}^{2}+(F_{c}{)}^{2}}\sqrt{(V_{f}-\frac{dx}{dt}{)}^{2}+{\frac{dy}{dt}}^{2}+(V_{c}-\frac{dz}{dt}{)}^{2}}$$
where $\frac{dx}{dt},\frac{dy}{dt}.\frac{dz}{dt}$—the rate of deformation movements of the tool. The equation describing the thermodynamics of the cutting process is given below [14]:(6)$$T_{1}T_{2}\frac{d^{2}Q_{z}}{dt^{2}}+(T_{1}+T_{2})\frac{dQ_{z}}{dt}+Q_{z}=kN$$
where $T_{1}=\frac{\lambda }{\alpha_{1}}$, $T_{2}=\frac{T_{h}}{\alpha_{2}}=\frac{h_{3}}{V_{c}\alpha_{2}}$—time constants of the thermodynamic subsystem, λ—the coefficient of thermal conductivity of the processed material, *h*_3_—the amount of tool wear along the back face, $k=\frac{k_{Q}\lambda h_{3}}{\alpha_{1}\alpha_{2}V_{c}}$—the transmission coefficient, α1—a scaling parameter of dimension, and α_2_—a dimensionless scaling parameter.

Taking into account the dependance of the proposed equation for the reaction forces, as well as relying on the approach for modeling the dynamics of the deformation movement of the tool used in the scientific school of Zakorotny V.L. [12], we assume that the model of deformations of the tool tip will take the following formula:(7)$$\left\{\begin{array}{c} \begin{array}{c} m\frac{d^{2}x}{dt^{2}}+h_{11}\frac{dx}{dt}+h_{12}\frac{dy}{dt}+h_{13}\frac{dz}{dt}+c_{11}x+c_{12}y+c_{13}z=F_{f}\\ m\frac{d^{2}y}{dt^{2}}+h_{21}\frac{dx}{dt}+h_{22}\frac{dy}{dt}+h_{23}\frac{dz}{dt}+c_{21}x+c_{22}y+c_{23}z=F_{p} \end{array}\\ m\frac{d^{2}z}{dt^{2}}+h_{31}\frac{dx}{dt}+h_{32}\frac{dy}{dt}+h_{33}\frac{dz}{dt}+c_{31}x+c_{32}y+c_{33}z=F_{c} \end{array}\right.$$
where m[$kg\cdot s^{2}/mm$]; h[kg⋅s/mm]; c[kg/mm]—matrices of inertia coefficients, dissipation coefficients, and stiffness coefficients, respectively.

As mentioned earlier, the formation of a wear site along the back face of a tool significantly affects the overall force response from the cutting process to the shaping movements of the tool. It is convenient to consider the force formed on the back face as:(8)$$F_{h}=(\sigma_{0}+k_{Q}^{F}Q_{h})h_{3}(t_{p}-y)e^{-K_{h}x}$$
where $\sigma_{0}$—compressive strength of the processed metal in [kg/mm], at zero degree contact temperature along the back face of the tool and the workpiece, $Q_{h}$, $k_{Q}^{F}$—the average coefficient of linear increase in the buoyant force with increasing contact temperature, and Kx—the coefficient describing the nonlinear increase in the pushing force when the tool and the workpiece approach.

Through the main angle in the plan, φ we decompose the force reaction on the x- and y-axes of deformation, as follows:(9)$$\left\{\begin{array}{c} {F_{h}}^{\left(x\right)}=\cos\varphi F_{h}\\ {F_{h}}^{\left(y\right)}=\sin\varphi F_{h} \end{array}\right.$$

The force reaction in the direction of the z coordinate is, in essence, nothing more than the friction force, which can be represented as:(10)$${F_{h}}^{\left(z\right)}=k_{t}F_{h}$$
where $k_{t}$—coefficient of friction.

The tool wear process is always bi-directional; one of the directions is focused on the addition of the cutting edge to the cutting process, as a result of which we observe the process of the contact area of the tool being formed along the back face, described in previous sections, through which the temperature field and force reaction are stabilized. The second direction is associated with an increase in the degrading features of the wear process, which subsequently leads to a significant change in the cutting properties and contact properties of the tool, and a sharp increase in the cutting force and associated vibrations. Based on these considerations, it is convenient to consider the kernel of an integral operator as the sum of two kernels:(11)$$h_{3}=\int_{0}^{A}(\beta_{1}e^{\alpha_{1}\left(\xi -A\right)}\;+\;\beta_{2}e^{\alpha_{2}\left(A-\xi \right)})d\xi ,$$
where ${(\beta }_{1}e^{\alpha_{1}\left(\xi -A\right)}+\beta_{2}e^{\alpha_{2}\left(A-\xi \right)})$—the sum of the kernels of the integral operator, $\beta_{1}e^{\alpha_{1}\left(\xi -A\right)}$—the core that determines the tool burn-in trajectory, $\beta_{2}e^{\alpha_{2}\left(A-\xi \right)}$ a the core determining the trajectory of degrading wear, and $\beta_{1}$, $\beta_{2}$, $\alpha_{1}$, $\alpha_{2}$,—parameters to be identified.

Thus, the basic version of the mathematical model of the digital twin is represented by the system of Equations (1)–(10), where expression (11) allows for calculating the current evaluation of the wear value of the cutting tool. Here, we note that the validation of the deterministic model of the digital twin proposed by us was carried out earlier and therefore is not given in this article. The results of a comparative analysis of experimental data and calculated model data were presented in a series of publications made by us earlier, the most informative of which are the following works: [18,19,20].

## 3. Description of the Hardware of the Vibration Monitoring System of the Cutting Process and the Results of the Experiment

For good operation in the digital twin system, the use of a system for monitoring the dynamics of the cutting process is required. The basis of such a system, as a rule, is a vibration diagnostic subsystem, which can be placed on the cutting tool itself, or rather on its holder [19]. It is also possible to consider the issues of measuring the temperature in the cutting area; however, this approach significantly complicates the entire vibration monitoring system. An example of such a system is the system shown in Figure 2. This system is based on an industrial general-purpose accelerometer of the IEPE standard (ICP) [21] with a built-in A603C01T charge converter amplifier, with the following specifications: Frequency range (+/−3 dB): 0.4–15,000 Hz. Sensitivity (+/−10%): 100 mB/g (10.2 mB/(m/s^2^)) and an ICP converter (IEPE), single channel, with a frequency range of 0.1–50,000 Hz. The frequency range of vibrations of the cutting tool tip, based on the results of previous studies [18,19,20], is in the range from 1 kHz to 5 kHz. According to the Nyquist–Shannon theorem, in order to restore such a signal from its discrete representation, the sampling frequency must be at least 2 times greater than the natural frequency of the original analog signal. Thus, the quantization frequency of the measured vibration acceleration signal will be 10 kHz. Based on these requirements, the E14-440 AD/DA ADC of the L-CARD campaign (manufacturer’s country is Russia) with the ability to transfer data via the USB 2.0 interface (USB Type B) was selected. The signal was measured for no more than 10 s, which ensured high reliability of the data stored in the experiment. The temperature regime corresponded to the requirements for ensuring the specified accuracy and reliability of measurements.

A steel shaft (steel 45) with a diameter of 75 cm was processed, cutting took place on a 1K625 machine, the processing mode was carried out at the speed of 124 m/min at a depth of 1 mm, and the feed rate was set to 0.11 mm per revolution.

As can be seen from Figure 2, the basis of the vibration monitoring system is piezoelectric vibration sensors for the cutting tool. Now however, new intelligent sensors have been widely developed, which have the ability to immediately digitally display vibration accelerations, speeds, and tool movements [22]. As a result, it is possible to develop an intelligent measurement system based on the use of digital meters, which must be installed on the instrument itself. However, data processing and strategic decision-making, such as decisions based on calculations of the residual strength of the cutting tool, cannot be carried out on the machine itself. Such solutions require large computing power, which can be found in the server part of the intelligent system for monitoring and predicting the dynamics of the cutting process. Modern technologies for data transmission and processing make it possible to do this rather quickly. Based on that, a promising vibration monitoring system should have two levels of control: operational, based on the primary processing of data coming from the instrument, and long-term strategic, which is associated with parallel processing modeling in a digital twin.

In the variant of a promising intelligent monitoring system presented in Figure 3, the digital twin is implemented on a cloud server, which receives data on the dynamics of the cutting process that has been pre-processed on an industrial computer installed on the machine itself. The strategic level implemented in the digital twin system allows you to calculate the current value of the wear of the cutting tool and predict a value for the remaining durability. Let us consider the operation of these two levels using one example of the flow, in which, observing the flow modes indicated above, we measured vibration accelerations, sequentially integrating them twice in a program written in the Matlab 2022b environment. The example of such data is shown below, in Figure 4, for the case measurements and calculations along the x-axis.

Similarly to the direction along the x-axis, measurements and calculations were carried out in the direction of the y-axis; the variant of such measurements and calculations is shown in Figure 5.

The vibrations measured and calculated in the direction of the z-axis are shown in Figure 6.

Thus, the above vibration monitoring system in its laboratory version allows for the measuring of vibration accelerations along the axes of tool deformations, as well as the calculation of the velocities and displacements of the tool tip in these directions.

## 4. Strategic and Operational Management Level

### 4.1. Strategic Level of Processing Management

Let us consider the strategic level of management; here, the issue of predicting the residual durability of a cutting tool is important, according to calculations carried out on the basis of modeling mathematical models in a digital twin system (see Figure 3). At the same time, it is necessary to take into account the actual path traveled by the tool, taking into account the vibrational movements of the top of the tool, which takes into account the processing of data obtained by the vibration monitoring system (see Figure 4, Figure 5 and Figure 6). Following that, the path traveled by the tool will be determined as the sum of the path *L*0, calculated from the elements of the CNC program (integral of the cutting speed and feed rate) and the virtual path traveled by the tool:(12)$$L=\sqrt{x^{2}+y^{2}+z^{2}}$$

The results of the systems of equations used in the modeling of the digital twin allow us to calculate the predicted development of the wear value of the cutting tool. Let us consider here that $\beta_{1}$, $\beta_{2}$, $\alpha_{1}$, $\alpha_{2}$—the parameters to be identified in the wear calculation model—were determined based on the results of preliminary experiments conducted by us earlier. The graph of the development of the cutting tool wear along the back face according to the calculated data are shown in Figure 7.

As can be seen from Figure 7, the development curve of cutting tool wear in the direction of the passed tool along the processed part passes through three areas:-the run-in area, where the preliminary wear of the cutting tool is formed;-the area of wear stabilization, where the amount of wear increases slowly enough;-the area of the formation of critical wear, in which the amount of wear of the cutting tool increases very quickly.

The most successful area for machining parts is the area of wear stabilization, as, in the area of critical wear, the cutting tool is non-operational. In strategic management, it is important for the machine operator to take into account the predicted residual durability of the cutting tool, which must be calculated using digital twin mathematical models using Equations (1)–(11). If the digital twin system shows that the value of the residual durability of the cutting tool is greater than the path that the cutting tool will take for the planned processing, then this operation is permissible; otherwise, the operator needs to replace the cutting tool.

### 4.2. Operational Level of Process Control

The actual wear of the cutting tool may be due to the influence of random factors, which are not possible to take into account in digital twin mathematical models. In other words, it is possible that a critical wear zone will be formed earlier than the calculated evaluation. Such a case of wear formation on a cutting tool often occurs in the case of heterogeneity of the processed material, or when changing the cutting mode (changing depth or feed), and it is also possible to influence external, disturbing factors such as the influence of chip formation properties, etc. To assess actual cutting tool wear, we conducted a separate experiment on the 1K625 machine, where the 45-steel (see Table 1) shaft was processed using the previously specified cutting parameters. As a tool, the holder MR TNR 2020 K11 and a five-sided plate 10,113–110,408 T15K6 were used, with the angle in the upper part (angle of attack) y0 = 35°, and the main angle in terms of φ = 80°.

To measure the wear of the cutting edge, at every step (a total of eight steps), the actual wear of the cutting edge on the back face was evaluated using a metallographic inverted labometer microscope option 4 with 5/20 wide-angle lenses with a linear field of view of 20 mm and with a digital camera for Ucam-1400 microscopes with a 1.4 µm × 1.4 µm matrix. The microscope itself and the measurement results are shown in Figure 8.

It is convenient to present the measurement results in the form of a table, which is given below (see Table 2).

It is convenient to interpret the measurement results of the wear of the cutting tool using a graphic, which is carried out below.

A comparative analysis of Figure 7 and Figure 9 shows that the critical wear zone begins, in the case of Figure 9, earlier than in Figure 7. This discrepancy is due to the chip formation process; during processing, we deliberately allowed cases of cutting with strong vibrations due to the accumulation of drain chips. As a result of these vibrations, there were changes in the contact between the back face of the tool and the workpiece, which led to the appearance of new elements in the wear zone. Such changes in the contact area of the tool and the workpiece are reflected in the signals measured by the vibration monitoring system, which can be estimated by calculating the RMS value of the signals (see expression (13)).
(13)$$\left\{\begin{array}{c} VA(\frac{d^{2}x}{dt^{2}})=\sqrt{\frac{1}{\tau }\int_{0}^{\tau }(\frac{d^{2}x}{dt^{2}})dt}\\ VA(\frac{d^{2}y}{dt^{2}})=\sqrt{\frac{1}{\tau }\int_{0}^{\tau }(\frac{d^{2}y}{dt^{2}})dt}\\ VA(\frac{d^{2}z}{dt^{2}})=\sqrt{\frac{1}{\tau }\int_{0}^{\tau }(\frac{d^{2}z}{dt^{2}})dt} \end{array}\right.$$

By calculating the RMS values of vibration accelerations, it is possible to determine peak fluctuations in the sample that exceed the calculated average value for the amplitude. After that, we will calculate an average value by the maximum amplitude of such oscillations:(14)$${\bar{a}}_{max}=\frac{1}{m}\sum_{j=1}^{m}\left|a_{j}\right|,\;\mathrm{for}\;\left|a_{j}\right|>\bar{a}$$
where $a_{j}$—the instantaneous value of vibrational acceleration at a discrete point in time, in mm/s^2^, and *m*—the sample size of the vibration acceleration values that go beyond the $\pm \bar{a}=VA$ calculated average values of vibrations. The results of such processing of the signals measured by the vibration monitoring system, for measurement point 1 of the curve in Figure 9, are shown below.

In Figure 10, the red dots indicate the peak values of vibration signals exceeding the calculated RMS value, and the green line shows the average level of peak values in total with the RMS value. The ratio of these two calculated values shows the degree of entropy of the measured signal:(15)$$\varepsilon =\frac{{\bar{a}}_{max}}{\bar{a}}$$

If the entropy index of the measured signal ε tends towards zero, then the entropy of the signals measured by the vibration monitoring system tends towards a maximum and vice versa, if the value of ε moves away from zero, then the information content of the signals captured by the vibration monitoring system increases. The results of calculating this indicator for three points of the wear curve, indicated in Figure 9 by arrows, are shown in Table 3.

As can be seen from Table 3, in the vicinity of the transition point of the wear curve of the cutting tool to the area of critical wear, the entropy of the measured signal increases significantly, which means that new formations will appear at the contact site of the rear face of the cutting tool and the workpiece. To illustrate this effect, we plot these three curves on one graph (see Figure 11).

As can be seen from Figure 11, at point 2, which characterizes the transition to the zone of critical wear on the graph of the actually measured wear (see Figure 9) of the cutting tool, there is indeed a significant increase in the entropy index. The most informative signal here is the signal taken from the vibration transducer mounted in the direction of the y-axis (see Figure 1). The signal taken from the vibration transducer mounted in the x-axis direction is somewhat less informative, and the signal along the z-axis is not informative enough.

Considering this, the operational control circuit can use the data captured by the vibration monitoring system and, if the entropy indicator along the x- and y-axes exceeds a certain predetermined value, signal to the operator of the metal-cutting machine that the wear curve of the cutting tool is rapidly moving into the zone of critical wear.

## 5. Discussion and Conclusions

The aim of this work was to make a conceptual scheme of a cutting process control system based on the joint use of a digital twin system and a vibration monitoring system for the cutting process. As a result, we have identified two contours of this system, the contour of strategic management and the contour of operational management. The conducted research has shown that the use of algorithms for predicting the value of residual durability of a cutting tool based on the use of deterministic virtual models is necessary but insufficient control tool. A necessary addition is the building of an operational control loop that allows you to refine the forecast of the development of wear of the cutting tool, proposed by solutions developed by virtual models.

The new approach proposed in this work allows us to combine the visibility and readability of deterministic digital twin mathematical models with the ability to recognize complex, noisy signals. Further development of the cutting control system from the point of view of the operational contour, as we see in the complication of the mathematical description for recognizing the origin point of critical wear zones due to increases in the number of informative signs in the measured signals, and the neural network processing of these data.

## Figures and Tables

**Figure 1 sensors-24-07403-f001:**
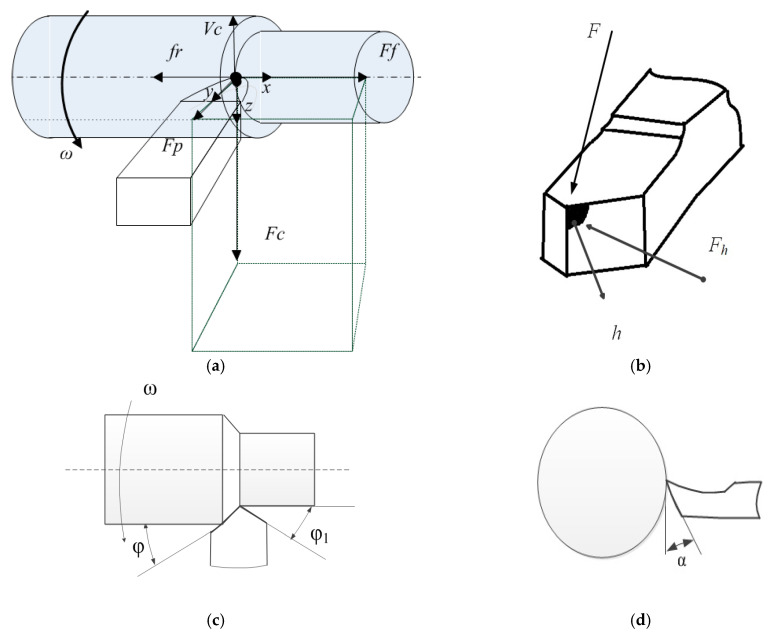
(**a**) Diagram of reaction forces and deformation axes, (**b**) directions of action forces, (**c**) main and auxiliary angles in the plan, (**d**) angle on the back surface.

**Figure 2 sensors-24-07403-f002:**
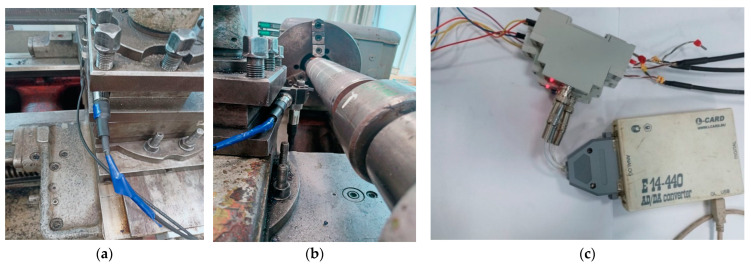
Vibration monitoring system on 1K625 machine, (**a**,**b**)—industrial accelerometers, (**c**)—amplifier converter and ADC.

**Figure 3 sensors-24-07403-f003:**
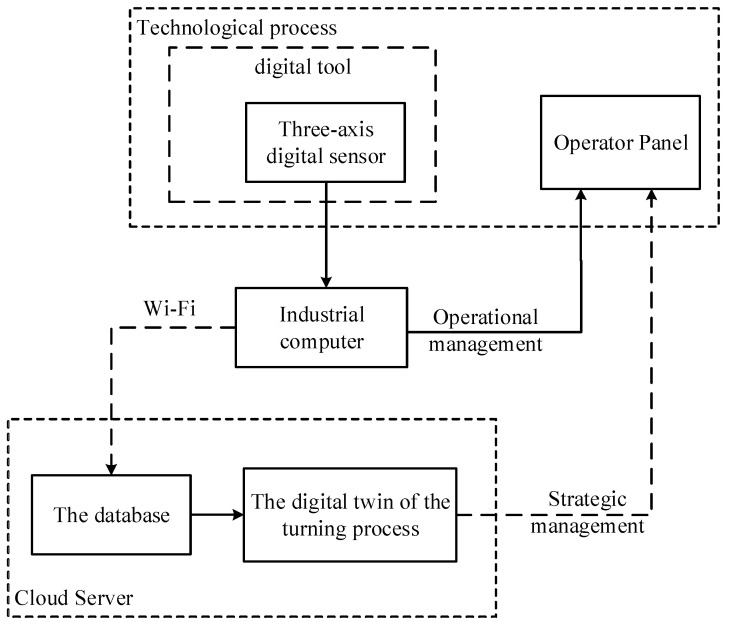
A promising intelligent vibration monitoring system.

**Figure 4 sensors-24-07403-f004:**
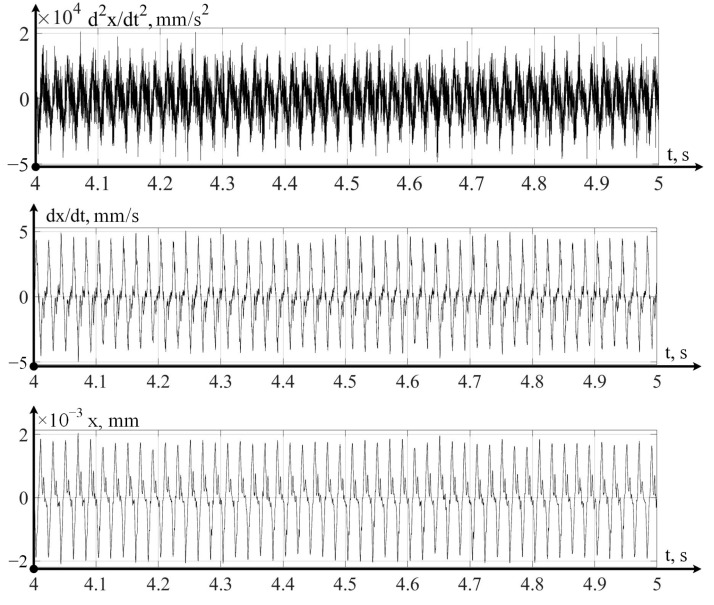
Vibrations of the cutting tool in the x-axis direction.

**Figure 5 sensors-24-07403-f005:**
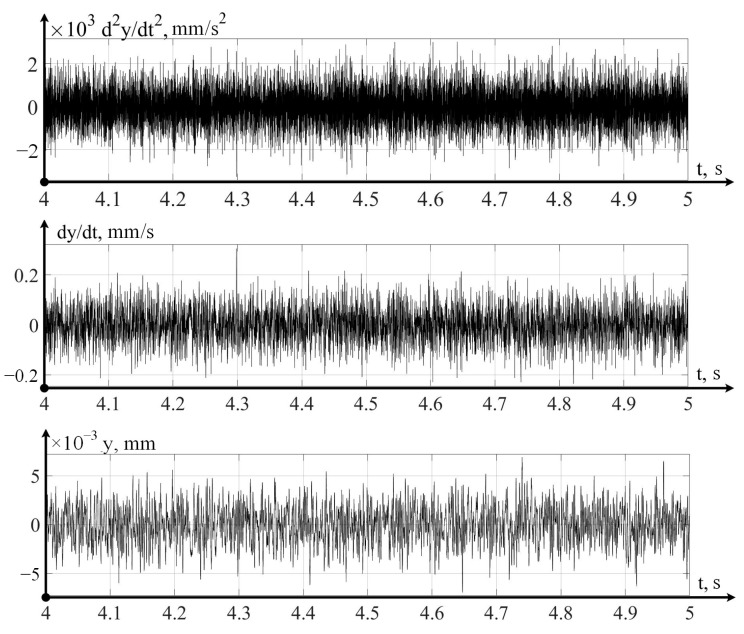
Vibrations of the cutting tool in the direction of the y-axis.

**Figure 6 sensors-24-07403-f006:**
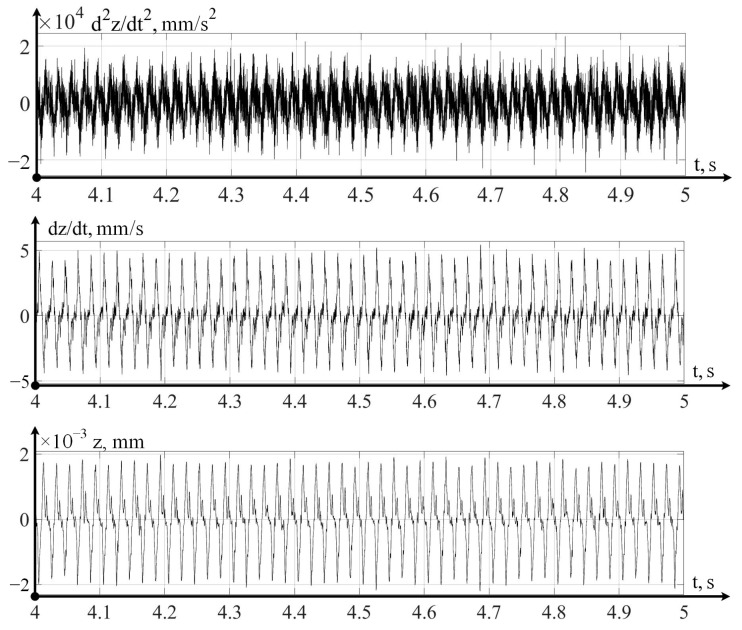
Vibrations of the cutting tool in the direction of the z-axis.

**Figure 7 sensors-24-07403-f007:**
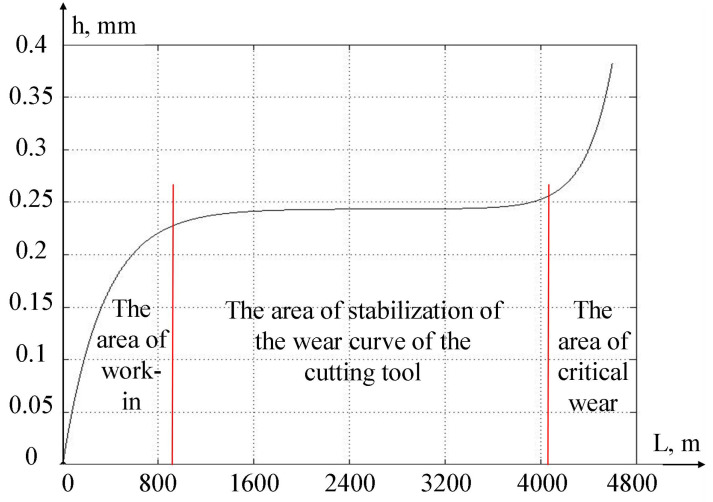
The wear curve.

**Figure 8 sensors-24-07403-f008:**
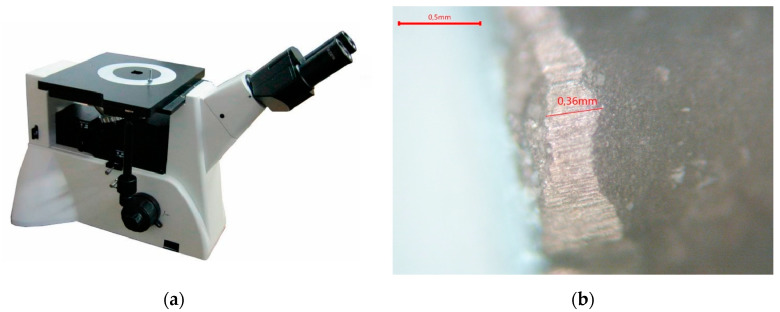
The appearance of the microscope (**a**), and one of the photographs obtained on it (**b**).

**Figure 9 sensors-24-07403-f009:**
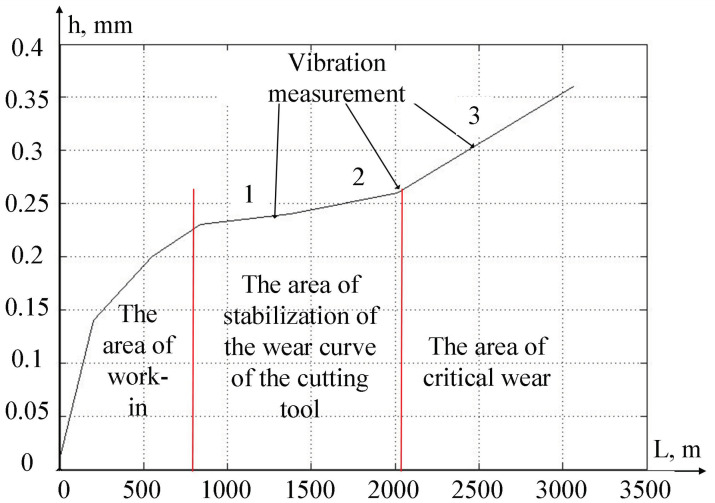
Measured wear curve of the cutting tool.

**Figure 10 sensors-24-07403-f010:**
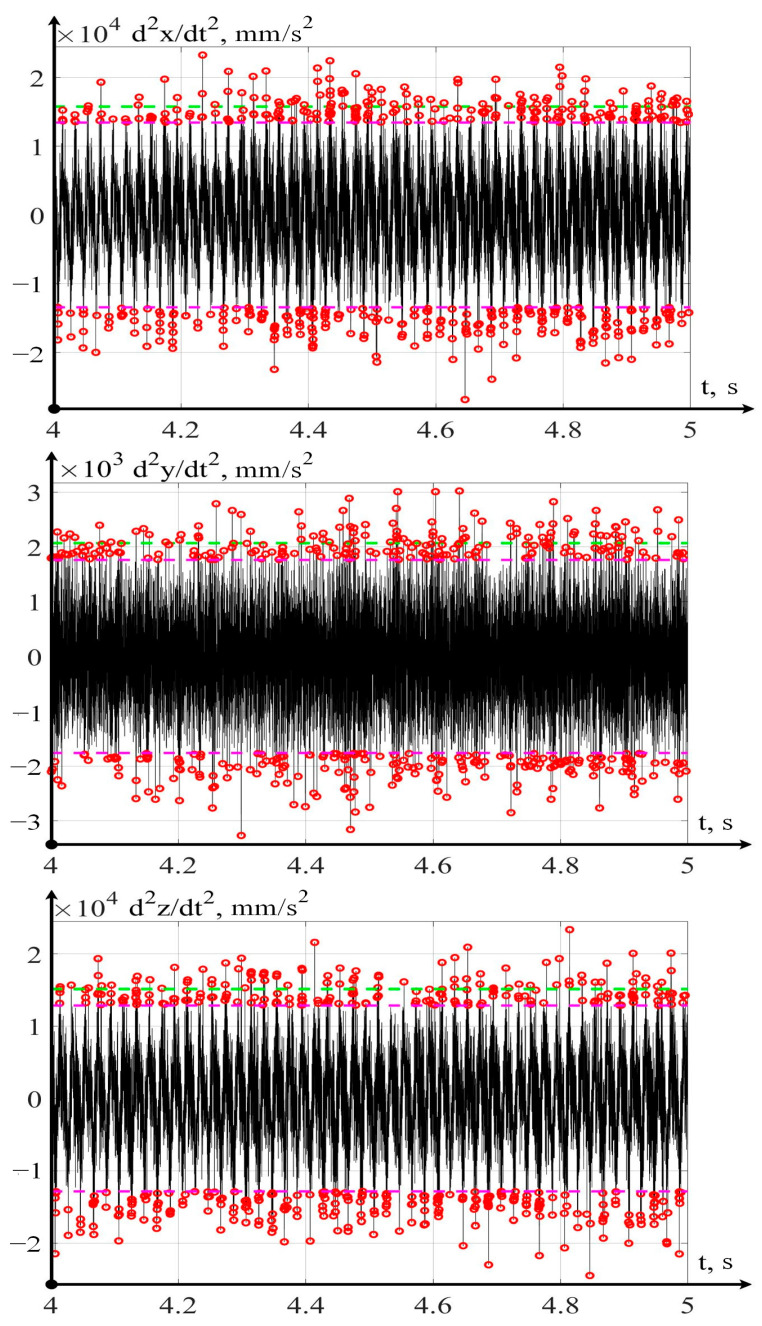
Example of calculating the average maxima of amplitude oscillations.

**Figure 11 sensors-24-07403-f011:**
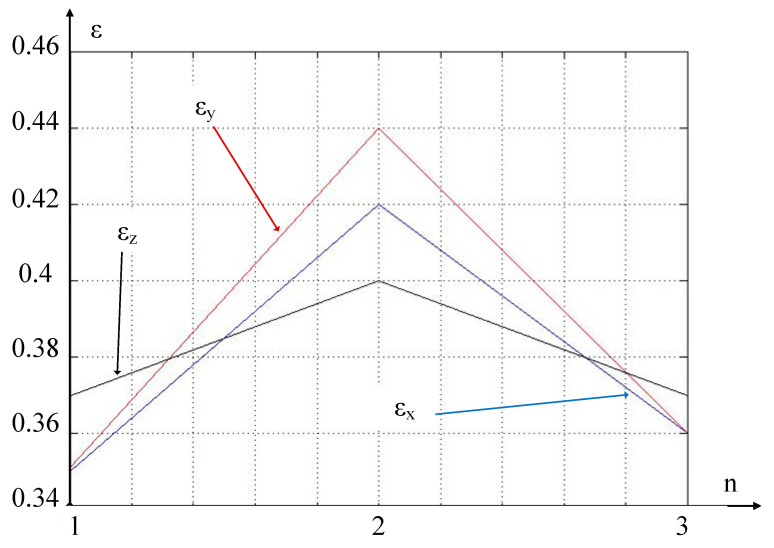
Results of calculation of entropy indicators.

**Table 1 sensors-24-07403-t001:** Chemical composition steel 45 [23,24,25].

C	Si	Mn	Cr	Ni	Ti	S	P
0.42–0.50	0.1–0.37	0.50–0.80	0.03	0.035	0.25	0.3	0.3

**Table 2 sensors-24-07403-t002:** Data on tool wear during processing.

The Path Traveled by the Cutter, m	Wear of the Cutting Plate, mm
0	0.01
202	0.11
552	0.20
840	0.23
1375	0.24
2010	0.26
3061	0.36

**Table 3 sensors-24-07403-t003:** Indicator of the entropy of vibration acceleration signals.

Signal	ε
d^2^x/dt^2^	0.35 0.42 0.36
d^2^y/dt^2^	0.351 0.44 0.36
d^2^z/dt^2^	0.37 0.4 0.37

## Data Availability

The datasets used and/or analyzed in the course of the current study are available from the relevant author upon reasonable request.

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
