# Peer review of "Development of a Conceptual Scheme for Controlling Tool Wear During Cutting, Based on the Interaction of Virtual Models of a Digital Twin and a Vibration Monitoring System"

_sensors, 2024, doi:10.3390/s24227403_

Round 1
Reviewer 1 Report
Comments and Suggestions for Authors
The present paper makes a conceptual scheme of a cutting process control system based on the joint use of a digital twin system and a vibration monitoring system for the cutting process. It is intresting, but the written of the paper is poor. The following issues should be noticed.
1. The article describes the hardware composition and certain performance parameters of the vibration monitoring system; however, it lacks a detailed analysis of the overall accuracy and reliability assessment of the system. For instance, there is no discussion on the measurement error range under different operating environments (temperature, humidity, vibration, etc.) or data regarding the system's stability over long-term operation.
2. In the tool wear prediction model, the author considers various factors affecting tool wear, but there is no discussion on the role of cutting fluids during the cutting process. We recommend that the author analyze the impact of cutting fluids within the context of the tool wear management system studied in this paper and consider whether related factors of cutting fluids should be further integrated into the model.
3. The author should pay attention to the font of the relevant variables in Figure 1; should they be italicized?
4. Should the number of relevant references be increased?
5. Should the conclusion and discussion sections be refined, and should the quality of English expression be improved?
Author Response
First of all, we are very grateful for your help in the design of our work.
Answers to questions:
- The article describes the hardware composition and certain performance parameters of the vibration monitoring system; however, it lacks a detailed analysis of the overall accuracy and reliability assessment of the system. For instance, there is no discussion on the measurement error range under different operating environments (temperature, humidity, vibration, etc.) or data regarding the system's stability over long-term operation.
Answer: We have clarified the information about the composition and characteristics of the measuring subsystem:
- This system is based on an industrial general-purpose accelerometer of the IEPE standard (ICP) with a built-in A603C01T charge converter amplifier. Frequency range (+/- 3dB): 0,4- 15 000 Hz. Sensitivity (+/- 10%): 100 mВ/g (10,2 mВ/(m/s2)) and the converter ICP (IEPE) single channel with frequency range 0,1- 50 000 Hz. The frequency range of vibrations of the cutting tool tip, based on the results of previous studies [14-16], is in the range from 1 kHz to 5 kHz. According to the Nyquist-Shannon theorem, in order to restore such a signal from its discrete representation, the sampling frequency must be at least 2 times greater than the natural frequency of the original analog signal. Thus, the quantization frequency of the measured vibration acceleration signal will be 10 kHz. Based on these requirements, the E14-440 AD/DA ADC of the L-CARD campaign (manufacturer country is Russia) with the ability to transfer data via the USB 2.0 interface (USB Type B) was selected. The signal was measured for no more than 10 seconds, which ensured high reliability of the data stored in the experiment. The temperature regime corresponded to the requirements for ensuring the specified accuracy and reliability of measurements.
- In the tool wear prediction model, the author considers various factors affecting tool wear, but there is no discussion on the role of cutting fluids during the cutting process. We recommend that the author analyze the impact of cutting fluids within the context of the tool wear management system studied in this paper and consider whether related factors of cutting fluids should be further integrated into the model.
Answer: The influence of lubricants in the cutting process on the evolutionary dynamics of the cutting process has not really been carried out by us. In the article, we limit ourselves to the dry cutting option. However, we plan to complicate the models of the digital twin, including by taking into account the influence of lubricants, in the near future.
- The author should pay attention to the font of the relevant variables in Figure 1; should they be italicized?
Answer: Thanks for your comment, Figure 1 has been clarified.
- Should the number of relevant references be increased?
Answer: The bibliographic list has been updated.
- Should the conclusion and discussion sections be refined, and should the quality of English expression be improved?
Answer: The article was additionally proofread by an English language specialist.
We thank you once again for your contribution to our research.

Reviewer 2 Report
Comments and Suggestions for Authors
Dear Authors,
The presented work is of great practical importance of tool wear management system for metalworking based on interaction between digital cutting process twin and vibration monitoring data. The work is a good analysis of experimental results during cutting. This work is probably can't publishable now, there exist a number of issues to be addressed prior to publication.
Specific Comments
1. The abstract is presented without any specific information. Only general phrases are presented. The application of neural networks to processing cutting information does not justify the novelty of the work. Please include the necessary information.
2. The abstract does not provide information on vibration signals and methods of their control.
3. The title of the article does not provide specific information about the article.
4. Introduction does not include the formation of the article's problems. There is no scientific justification for the proposed study. Please check.
5. Figure 1, the diagram of the forces during turning is shown in a very simplified form. Please correct.
6. Where did formulas 1-6, 9-11 come from?? If they were derived by the authors, it is necessary to include a justification for the calculations, otherwise it is necessary to indicate the source. Please clarify.
7. The formatting of Figure 2 is broken. Please correct it.
8. Lines 145-187, the description of the experiment does not provide data on the tool, part material, equipment. Please describe it.
9. Lines 277-306, it is necessary to justify the selection of results of the characteristics of the vibro-acoustic signal, frequency ranges. The signal from the sensor may have its own errors. That is, the absolute value of the signal may not describe the cutting process in reality. There are also types of workpiece material during processing, in which an increase in the signal can serve as an improvement in cutting quality and an increase in chip formation. It is necessary to prove the correlation of the signal and wear. Please clarify these issues.
10. The article essentiall "Discussion section". Supplement the content of the manuscript.
Author Response
First of all, we are very grateful for your help in the design of our work.
Answers to questions:
- The abstract is presented without any specific information. Only general phrases are presented. The application of neural networks to processing cutting information does not justify the novelty of the work. Please include the necessary information.
Answer: The abstract has been completely redesigned
- The abstract does not provide information on vibration signals and methods of their control.
Answer: Changes have been made to the annotation.
- The title of the article does not provide specific information about the article.
Answer: We have changed the title of the article
- Introduction does not include the formation of the article's problems. There is no scientific justification for the proposed study. Please check.
Answer: We have made appropriate additions to the introduction.
- Figure 1, the diagram of the forces during turning is shown in a very simplified form. Please correct
Figure 1 is expanded, here we tried to reveal the features of the formation of a force reaction
- Where did formulas 1-6, 9-11 come from?? If they were derived by the authors, it is necessary to include a justification for the calculations, otherwise it is necessary to indicate the source. Please clarify.
Answer: The corresponding expressions were developed by a team of authors, a list of publications revealing a more detailed synthesis of these expressions is given in our article: “The results of a comparative analysis of experimental data and calculated model data are presented in a series of publications made by us earlier, the most informative of them are the following works [14-16].”
- The formatting of Figure 2 is broken. Please correct it
Answer: The picture is formatted.
- Lines 145-187, the description of the experiment does not provide data on the tool, part material, equipment. Please describe it.
Answer: Data added.
- Lines 277-306, it is necessary to justify the selection of results of the characteristics of the vibro-acoustic signal, frequency ranges. The signal from the sensor may have its own errors. That is, the absolute value of the signal may not describe the cutting process in reality. There are also types of workpiece material during processing, in which an increase in the signal can serve as an improvement in cutting quality and an increase in chip formation. It is necessary to prove the correlation of the signal and wear. Please clarify these issues.
Answer: In the article, we tried to justify the choice of the results of the characteristics of the vibroacoustic signal and frequency ranges in as much detail as possible. We have added additional data reflecting the answer to this question.
- The article essentiall "Discussion section". Supplement the content of the manuscript
Answer: We focus on the discussion of the results in each section of the article, and we present the main conclusions in conclusion. We consider it inappropriate to repeat this material in a separate section.
We thank you once again for your contribution to our research.

Reviewer 3 Report
Comments and Suggestions for Authors
The manuscript presents an interesting topic on the use of digital twin technology for machining control systems. However, several significant issues in content structure, technical details, and language need to be addressed before the work can be considered for publication.
1. Repetition of Ideas
- Page 1, Lines 42-43: There is a repetition of the same concept in this paragraph. The sentences "One of the ways to solve this problem is to use a new digital paradigm in quality management and control systems, which has been called a digital twin [5-7]" and "The solution to these problems is possible through the use of digital twin technology [8-9]" convey the same idea. It is recommended to merge or rephrase these statements to avoid redundancy and improve clarity.
2. Use of Superlatives
- Page 1, Lines 45-46: The use of superlative forms should be avoided in scientific writing. The phrase "Y. Altintas, who is today the most famous specialist in the field of digital twins of metalworking control systems,..." is subjective and inappropriate for an academic context. A more neutral description would be preferred, such as "a leading specialist" or "a prominent figure in the field."
3. Global Perspective
- Page 2, Line 52: The paper should be addressed to the global scientific community and not limited to a specific country or region. The sentence "In Russia, the technology of building digital twins in terms..." restricts the scope of the study to Russia. It is recommended to present the technology from a global point of view, highlighting its broader relevance and application worldwide.
4. Terminology
- Page 2, Line 73: The term "front surface" is not commonly used in the context of machining. It would be more appropriate to replace it with the standard term "rake face."
5. Missing Parameters in Figure 1
- Page 3, Lines 93-95: The authors reference Figure 1 to present parameters for equations 3 and 4 but the mentioned parameters are not included in the figure. This is a significant oversight that must be corrected. Figures should provide all the relevant information needed to understand the associated equations.
6. Misuse of "First Chapter"
- Page 3, Line 103: The phrase "first chapter" is incorrect in a scientific paper. Instead, the appropriate term would be "previous section" or "earlier section." Scientific papers do not typically refer to chapters.
7. Mathematical Errors
- Page 8, Equation 12: The equation contains an error, as it incorrectly uses (x²) twice under the square root. The authors should carefully review and correct the mathematical formulation.
8. Validation of Mathematical Model
- The paper lacks a proper validation of the mathematical model for cutting forces. Validation is critical for ensuring the accuracy and reliability of the proposed model. This omission is a major flaw in the study.
9. Missing Experimental Boundary Conditions
- The authors do not provide any experimental boundary conditions for their tests. Without this information, it is impossible to assess the relevance and applicability of the experimental results. Boundary conditions are essential for replicating and validating the study's findings.
10. Limited Tool Wear Analysis
- The use of only one tool wear evolution in the study is insufficient to validate the model robustly. A comprehensive tool wear analysis should involve multiple scenarios and cutting conditions to ensure the model's robustness.
11. Lack of Dependency on Tool-Workpiece Geometry
- The developed models do not account for the geometry of the tool and workpiece. This is a critical limitation, as changes in the geometrical parameters of the cutting tool could render the model invalid. The authors need to address this by making the model geometry-dependent.
Final Recommendation: Due to the issues mentioned above, including repetition of ideas, improper validation, lack of boundary conditions, terminology inaccuracies, and poor language quality, I recommend rejecting this manuscript.
Comments on the Quality of English Language
Language and Terminology Issues
- The paper suffers from poor English language quality and the use of uncommon nomenclature that is not standard in the field. The manuscript would benefit greatly from a thorough language revision to improve clarity and adherence to standard terminology.
Author Response
First of all, we are very grateful for your help in the design of our work.
Answers to questions:
- Repetition of Ideas: Page 1, Lines 42-43: There is a repetition of the same concept in this paragraph. The sentences "One of the ways to solve this problem is to use a new digital paradigm in quality management and control systems, which has been called a digital twin [5-7]" and "The solution to these problems is possible through the use of digital twin technology [8-9]" convey the same idea. It is recommended to merge or rephrase these statements to avoid redundancy and improve clarity.
- Use of Superlatives: Page 1, Lines 45-46: The use of superlative forms should be avoided in scientific writing. The phrase "Y. Altintas, who is today the most famous specialist in the field of digital twins of metalworking control systems,..." is subjective and inappropriate for an academic context. A more neutral description would be preferred, such as "a leading specialist" or "a prominent figure in the field."
- Global Perspective: Page 2, Line 52: The paper should be addressed to the global scientific community and not limited to a specific country or region. The sentence "In Russia, the technology of building digital twins in terms..." restricts the scope of the study to Russia. It is recommended to present the technology from a global point of view, highlighting its broader relevance and application worldwide.
Answer: The introduction has been completely rewritten in accordance with your recommendations.
- Terminology: Page 2, Line 73: The term "front surface" is not commonly used in the context of machining. It would be more appropriate to replace it with the standard term "rake face."
Answer: The corresponding changes have been made to the article.
- Missing Parameters in Figure 1: Page 3, Lines 93-95: The authors reference Figure 1 to present parameters for equations 3 and 4 but the mentioned parameters are not included in the figure. This is a significant oversight that must be corrected. Figures should provide all the relevant information needed to understand the associated equations.
Answer: The corresponding changes have been made to the article.
- Misuse of "First Chapter": Page 3, Line 103: The phrase "first chapter" is incorrect in a scientific paper. Instead, the appropriate term would be "previous section" or "earlier section." Scientific papers do not typically refer to chapters.
Answer: Thank you for this remark, this is clearly our mistake, appropriate adjustments have been made to the material of the article.
- Mathematical Errors: Page 8, Equation 12: The equation contains an error, as it incorrectly uses (x²) twice under the square root. The authors should carefully review and correct the mathematical formulation.
Answer: Thank you for this remark, this is clearly our mistake, appropriate adjustments have been made to the material of the article.
- Validation of Mathematical Model: The paper lacks a proper validation of the mathematical model for cutting forces. Validation is critical for ensuring the accuracy and reliability of the proposed model. This omission is a major flaw in the study.
Answer: The presented work is part of an extensive study aimed at synthesizing the structure of virtual models of digital counterparts of metal-cutting machines. As part of this work, we solve various problems, the solution of one of which is presented in this article. Validation of our proposed models was carried out by us earlier and its results have been published in a number of our previous publications. In the article we provide links to these publications of ours: “The results of a comparative analysis of experimental data and calculated model data are presented in a series of publications made by us earlier, the most informative of them are the following works [14-16].”
- Missing Experimental Boundary Conditions: The authors do not provide any experimental boundary conditions for their tests. Without this information, it is impossible to assess the relevance and applicability of the experimental results. Boundary conditions are essential for replicating and validating the study's findings.
- Limited Tool Wear Analysis: The use of only one tool wear evolution in the study is insufficient to validate the model robustly. A comprehensive tool wear analysis should involve multiple scenarios and cutting conditions to ensure the model's robustness.
Answer: In the work, we provided information about the parameters of metal cutting on the appropriate metal cutting machine (cutting speed, depth and feed). Here we also indicated which metal we processed - we further clarified this issue. We have described in detail the measuring subsystem, its composition and characteristics. The graphs show the general cutting path, etc. However, it should be clarified here that the experimental results we have obtained have their own unique feature. This feature is due to the fact that we cannot specify the quality level of the metal cutting machine, we cannot take into account dislocations in the processed material and the unique properties of the cutting plate. All this together indicates the random nature of the wear process of the cutting tool. This is the idea that underlies our work. We are aware that no matter how complex a model of a digital twin system can accurately predict the development of wear during the cutting process. The logic of the article is that we complement this system with another subsystem that protects the machine from extreme situations associated with a deviation of real wear from its predicted value. Therefore, an experiment on a real machine is just a way to show the relevance of such a solution.
- Lack of Dependency on Tool-Workpiece Geometry: The developed models do not account for the geometry of the tool and workpiece. This is a critical limitation, as changes in the geometrical parameters of the cutting tool could render the model invalid. The authors need to address this by making the model geometry-dependent.
Answer: Thank you for this question. This question overlaps with some of your previous questions, in particular the question of validating the force reaction model. It is appropriate to repeat here that the model itself and its parametric content are described in more detail in a series of our previous works. However, we clarified in the materials of the article the parameters that reveal the influence of the geometry of the cutting wedge on the formation of a force reaction:
- where the coefficients of decomposition of the cutting force F on the axis of the tool deformation, these coefficients depend on the angles indicated in Figure 1 in the tool plan (φ,φ1,α), it is here the geometry of the cutting plate is taken into account [4].
- where p- the constant that determines the valuation of the specific chip pressure, per millimeter of the area of the layer cut during cutting, the geometry of the cuttingplatealsoplays a roleinthis
We thank you once again for your contribution to our research.

Reviewer 4 Report
Comments and Suggestions for Authors
This manuscript presents a interesting topic, and its contributions is significant. Below are specific comments for consideration:
1. In page 3, Fh in Eq. (1) is not noted.
2. In line 105, page 3, Ff, Fp, Fc are not needed to note again.
3. In line 112, page 4, what is the meaning of “с2/м3”?
4. In line 125, page 5, “Through the main angle in the plan – φ, …” must be illustrated or explained.
5. In line 138, page 5, where is “figure (12)”?
6. In page 8, please examine Eq. (12)
7. In line 217, page 8, how can we get “the calculated data”?
8. In figure 7 and 9, the expression of “The area of work-in” does not match to those of the other areas. It is about wear. And please examine “run-in” in line 222, page 9, vs “work-in” in figure 7.
9. Please check eq. (13)
10. Please consider how to embody the bi-directional process of “twin” in this research?
11. In eq. (14), what is the
represent?

Author Response
First of all, we are very grateful for your help in the design of our work.
Answers to questions:
- In page 3, Fh in Eq. (1) is not noted.
Answer: Appropriate changes have been made.
- In line 105, page 3, Ff, Fp, Fc are not needed to note again.
Answer: Appropriate changes have been made.
- In line 112, page 4, what is the meaning of “с2/м3”?
Answer: This is a typo that we have corrected.
- In line 125, page 5, “Through the main angle in the plan – φ, …” must be illustrated or explained.
Answer: We gave the corresponding explanations in Figure 1.
- In line 138, page 5, where is “figure (12)”?
- In page 8, please examine Eq. (12)
- In line 217, page 8, how can we get “the calculated data”?
Answer: We have taken into account all the questions on mathematical models and made the necessary corrections.
- In figure 7 and 9, the expression of “The area of work-in” does not match to those of the other areas. It is about wear. And please examine “run-in” in line 222, page 9, vs “work-in” in figure 7.
Answer: However, it should be clarified here that the experimental results we have obtained have their own unique feature. This feature is due to the fact that we cannot specify the quality level of the metal cutting machine, we cannot take into account dislocations in the processed material and the unique properties of the cutting plate. All this together indicates the random nature of the wear process of the cutting tool. This is the idea that underlies our work. We are aware that no matter how complex a model of a digital twin system can accurately predict the development of wear during the cutting process. The logic of the article is that we complement this system with another subsystem that protects the machine from extreme situations associated with a deviation of real wear from its predicted value. Therefore, an experiment on a real machine is just a way to show the relevance of such a solution.
- Please check eq. (13)
Answer: We have taken into account all the questions on mathematical models and made the necessary corrections.
- Please consider how to embody the bi-directional process of “twin” in this research?
Answer: This issue requires additional research, we have not tried to work on it yet.
- In eq. (14), what is the represent?
Answer: We have taken into account all the questions on mathematical models and made the necessary corrections.
We thank you once again for your contribution to our research.

Round 2
Reviewer 1 Report
Comments and Suggestions for Authors
It can be accepted in the present version.
Comments on the Quality of English Language
No comments.
Author Response
Thank you for your work on editing our article!
Reviewer 2 Report
Comments and Suggestions for Authors
Dear Authors,
Thanks to the authors for their time to improve the quality of their work. The level of work has been improved.
I think the authors have successfully made adjustments to comments 3 (the article “a” may need to be replaced with “the” in the context), comment 4 (the necessary section has been added), comment 5 (the figure has been supplemented), comment 7 (the figure is correct), comment 8 (data has been entered, but market labeling has been introduced, which is not technical information).
The rest of the comments require additions (it is not clear what was), in particular, the article should have a «Discussion», this is a mandatory requirement for publications.
Overall, the level has been improved, but the manuscript needs to be improved. The publication of the article is possible at the discretion of the editors of the publication.
Author Response

(The authors gave the same response as above.)

Reviewer 3 Report
Comments and Suggestions for Authors
The authors have addressed all comments. Paper should be considered for publication.
Comments on the Quality of English Language
None
Author Response

(The authors gave the same response as above.)

Reviewer 4 Report
Comments and Suggestions for Authors
The manuscript is revised according to the reviewers' comments. But there are still several problems should be examined.
1. In line 110, page 3, ‘a)’ and 'b)' appear twice.
2. In line 182, page5, "...presented in figure (2):" follows Eq.(11). Is it right?
3. In line 269, page10, does 'L0' calculated by Eq.(12)?
4. Please examine Table 1, ',' or '.'?
Author Response
First of all, we thank you for your work in reviewing our article. According to all your comments, we have made changes to the work:
1) In line 110, page 3, ‘a)’ and 'b)' appear twice.
Answer: We have corrected this error, replaced the numbering according to the drawing.
2) In line 182, page5, "...presented in figure (2):" follows Eq.(11). Is it right?
Answer: This is an annoying typo, we have removed it.
3) In line 269, page10, does 'L0' calculated by Eq.(12)?
Answer: We have expanded the understanding of the word "calculated": That is, the path traveled by the tool will be determined as the sum of the path L0 calculated from the elements of the CNC program (integral of the cutting speed and feed rate) and the virtual path traveled by the tool.
4) Please examine Table 1, ',' or '.'?
Answer: We have replaced all commas with dots.